# Revealing real-time 3D in vivo pathogen dynamics in plants by label-free optical coherence tomography

Jos de Wit [1], Sebastian Tonn [2], Mon-Ray Shao[2], Guido Van den Ackerveken [2] & Jeroen Kalkman [1]

Microscopic imaging for studying plant-pathogen interactions is limited by its reliance on invasive histological techniques, like clearing and staining, or, for in vivo imaging, on complicated generation of transgenic pathogens. We present real-time 3D in vivo visualization of pathogen dynamics with label-free optical coherence tomography. Based on intrinsic signal fluctuations as tissue contrast we image filamentous pathogens and a nematode in vivo in 3D in plant tissue. We analyze 3D images of lettuce downy mildew infection (*Bremia lactucae*) to obtain hyphal volume and length in three different lettuce genotypes with different resistance levels showing the ability for precise (micro) phenotyping and quantification of the infection level. In addition, we demonstrate in vivo longitudinal imaging of the growth of individual pathogen (sub)structures with functional contrast on the pathogen micro-activity revealing pathogen vitality thereby opening a window on the underlying molecular processes.

3D live imaging of plant infection by filamentous microbes is crucial to understanding mechanisms of pathogen virulence, host disease susceptibility, and plant resistance[1–3]. Most of the current studies are destructive to the plant, thereby prohibiting longitudinal in vivo studies[4,5]. Non-destructive imaging only can be achieved using genetically transformed pathogens expressing fluorescent marker proteins that are traced by in vivo microscopy[1,6–9]. Engineering such pathogen transformants is complicated, has to be developed for every pathogen species or isolate, and results in genetically-modified organisms that require dedicated facilities for working with them[10]. Moreover, many pathogen species cannot easily be transformed, in particular most obligate biotrophs such as the downy mildews[11,12]. These drawbacks can be overcome by label-free imaging, such as based on intrinsic scattering[13] or absorption contrast. Label-free imaging can be applied to every pathogen, gives simultaneous structural information, and allows imaging in its native biological state over prolonged amounts of time. The impact of label-free imaging has been limited so far due to its inability to visualize pathogenic structures because of their small size, limited intrinsic optical contrast, and their embedding in highly scattering plant tissue. Conventional optical coherence tomography (OCT) provides label-free 3D imaging of plants deep inside the leaf[14], especially when combined with water infiltration[15]. However, it fails to distinguish between different types of equally scattering tissue, and thus cannot distinguish pathogenic structures from plant tissue. In contrast, dynamic OCT creates cell-specific contrast using sub-resolution motility inside the cells. The sub-resolution scatterer movements, which have been linked to metabolism[16] and mitochondrial activity[17], cause temporal fluctuations in the OCT scattering intensity. In dynamic OCT, the spectral content of these fluctuations is analyzed and translated to a false color, which provides label-free contrast images from tissue in its native state[18].

Here, we show that dynamic optical coherence tomography (dOCT)[18] enables high-contrast label-free in vivo 3D imaging of plant pathogens. We utilize the unique dynamic fingerprints of plant and pathogen tissue, water infiltration of leaves to reduce light scattering[15], and OCT's deep tissue imaging capabilities to create high-contrast images of pathogen structures deep in plant tissue. We demonstrate in vivo visualization of different plant hosts infected with downy

[1]Department of Imaging Physics, Delft University of Technology, Delft, The Netherlands. [2]Translational Plant Biology, Department of Biology, Utrecht University, Utrecht, The Netherlands. ✉e-mail: j.kalkman@tudelft.nl

mildew and nematode infection of roots. The potential for digital (micro) phenotyping is demonstrated by imaging and quantifying the 3D infection morphology of lettuce downy mildew (*Bremia lactucae*) in leaf tissue for three lettuce genotypes with different resistance levels. Through segmentation of *B. lactucae* hyphae in dOCT images we enable precise quantification of hyphal volume and length, which was in agreement with DNA-based quantification of whole leaf pathogen presence using qPCR[19]. Finally, we image and quantify individual longitudinal *B. lactucae* growth trajectories, thereby exploiting the non-destructive in-vivo imaging ability of dOCT in contrast to destructive methods. Our results show the feasibility of label-free longitudinal and quantitative plant infection studies. Their use for both quantitative digital phenotyping and fundamental biology studies can aid in the development of disease-resistant crops that are of paramount importance to feed the growing world population[20].

## Results

### Dynamic OCT imaging

In vivo label-free high-contrast dynamic OCT images were generated using spectral analysis of the temporal OCT signal[18], see Fig. 1a. To improve the imaging depth plant leaf disks were infiltrated with water[15] or perfluorodecalin[21]. Next, a series of B-scans is obtained at the same location over a time of 1.5 seconds. The fluctuations of the OCT amplitude in time characterize the sub-cellular tissue dynamics of each tissue type. Static structures give a stationary signal, while structures with strong sub-cellular motion, such as in *B. lactucae* hyphae, give a highly fluctuating signal. Second, the time scale of the motion is captured in the amplitude spectrum, which is obtained by Fourier transforming the time signal. The amplitude spectrum is divided into three bands that are optimized (Supplementary Fig. 1) for visualization of biological tissue: low (0 Hz), medium (0.7–4.8 Hz), and high frequency

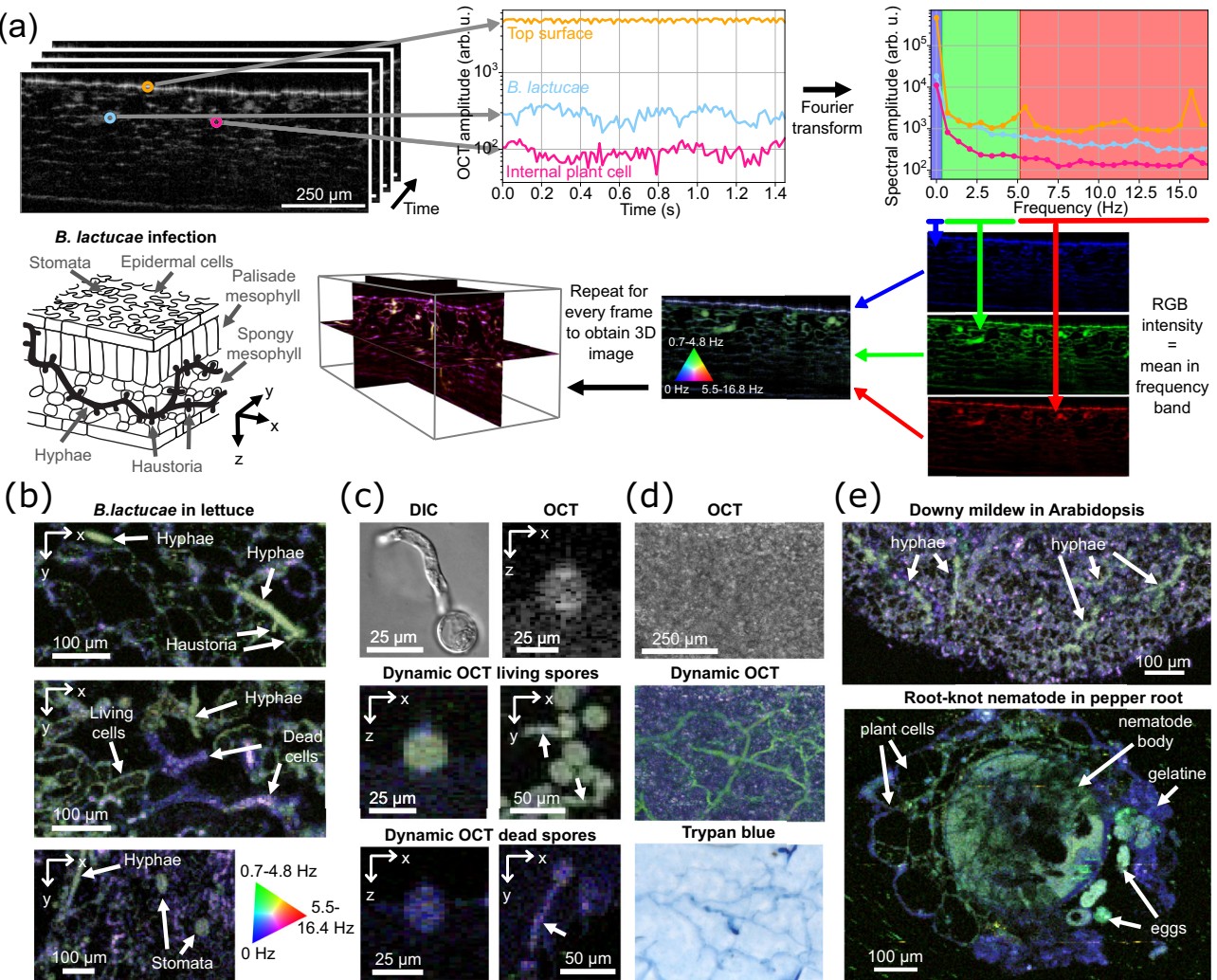

**Fig. 1 | Dynamic OCT (dOCT) imaging. a** Overview of the dOCT imaging process: a series of B-scans are acquired and, for each pixel, the amplitude over time is Fourier transformed to obtain the amplitude spectrum. The logarithm of the mean spectral amplitude in three frequency bands provides the intensity for the three colors resulting in an RGB false color dOCT image. Repeating this for parallel frames gives a volumetric image that captures the 3D structure of the hyphae, as also shown in the line drawing (which also indicates the *xyz* geometry). **b** Horizontal cross-section of dOCT images of *Bremia lactucae* infected lettuce leaves showing hyphae and haustoria invading spongy mesophyll cells (top), dead spongy mesophyll cells caused by plant resistance to *B. lactucae* infection (middle), epidermal cells, and plant stomata (bottom). The images are a maximum intensity projection (MIP) over 7 µm, 4 µm, and 1.4 µm respectively. **c** Comparison of differential interference contrast microscopy and dOCT on living and dead spores (examples from *N* > 10 individual living and dead spores). Supplementary movie 1 shows the motion of the scattering content and OCT speckle. Arrows indicate germ tubes. **d** MIP over the full axial range of the conventional OCT image, the dOCT image, and the Trypan blue stained brightfield microscopy image of the same sample demonstrating that dOCT can label-free image *B. lactucae* hyphae inside the plant leaf (example from *N* = 3 replicates). **e** Dynamic OCT images (horizontal cross-section, MIP over 4 µm depth) of downy mildew (*Hyaloperonospora arabidopsidis*) infection in *Arabidopsis* palisade mesophyll tissue and root-knot nematode (*Meloidogyne incognita*) infection in pepper plant roots. The conventional OCT reference images are included in Supplementary Fig. 8.

(5.5–16.4 Hz). The average amplitude in each band gives the color intensity value for the blue, green, and red channels respectively resulting in a false color image. The *B. lactucae* hyphae and plant cell outlines have different scattering amplitude spectra which gives them a different false color. Repeating this process at multiple B-scans results in a 3D image with dOCT contrast.

Different biological features can be observed from in vivo images of *B. lactucae*-infected leaf segments, see Fig. 1b. The detailed structure of a *B. lactucae* hyphae with two protruding haustoria (specialized structures that penetrate plant cells to suppress plant immunity and extract nutrients) is visible in the dOCT image (top panel). Also, dead plant cells, that result from hypersensitive cell death in a resistant lettuce genotype in response to an avirulent *B. lactucae* isolate[22,23], can be effectively imaged (middle panel). There is a clear contrast between living plant cells (green) and dead plant cells (purple). Stomata can be distinguished on the leaf epidermal layer (lower panel), with the stomata standing out in green in contrast with the blue-colored patches from the strong stationary reflection of the leaf top surface.

Evidence that the dOCT hyphal structures are linked to biological activity is demonstrated by a comparison of a differential interference contrast image with dOCT, see Fig. 1c. A differential interference contrast brightfield image shows the motility of the scattering cellular content in a (germinating) spore (Supplementary movie 1) which causes the speckle fluctuations observed in real-time OCT B-scan images. The dOCT image shows the activity of the living spores and germ tubes (in green, see arrows). Heat-killed spores no longer showed dynamic scattering as the biological activity inside the (germinating) spores was disrupted. The remaining static speckle in OCT images resulted in a purple-colored dOCT image of the spores (blue from the static 0 Hz signal, red from high-frequency noise, not green). This confirms that the cellular activity inside the pathogen is the biological origin of the medium frequency dOCT contrast.

Confirmation that the bright green structures in the dOCT images are *B. lactucae* hyphae was obtained from a comparison with the same leaf sample, which was Trypan blue stained, cleared, and imaged in brightfield, see Fig. 1d. While the maximum intensity projection (MIP) of the conventional OCT image shows no sign of *B. lactucae* because it only shows scattering strength and not its fluctuations, in contrast, the MIP of the dOCT image clearly shows the pathogen, in even greater detail than the Trypan blue-stained brightfield image of the same area.

The dOCT contrast between plant and pathogen was also demonstrated in other plant-pathogen systems, see Fig. 1e. Dynamic OCT images of *Arabidopsis* leaves infected with downy mildew (*Hyaloperonospora arabidopsidis*) showed clear hyphae. Similar detection of downy mildew hyphae was obtained in radish (Supplementary Fig. 8). Also, images of root-knot nematodes (*Meloidogyne incognita*) in pepper root showed contrast between the nematode eggs, the gelatinous mass that holds them together, the plant cells, and the nematode body. These examples illustrate the wide potential for dOCT to study different plant-pathogen and plant-parasite systems, and explore different plant tissues.

## Micro-phenotyping with dynamic OCT

To evaluate the value of 3D label-free plant pathogen imaging for digital phenotyping, we quantified the infection level in the leaf tissue of three different lettuce varieties to downy mildew during the symptomless colonization of the leaf mesophyll tissue, i.e., before sporulation, Fig. 2a. Imaging was performed on a total of 48 volumetric leaf segments (12 plants per lettuce variety, 2 leaf disks per plant, 2 3D dOCT volumes per leaf disc).

Segmentation and quantification of *B. lactucae* colonization requires a high contrast. Despite the fact that both *B. lactucae* and plant leaf tissue are biologically active, the two organisms could be distinguished by optimizing a weighted combination of the signals

from the different colors (frequency bands), see Fig. 2b, Supplementary Fig. 3, and Supplementary section 1.3. The combination 0.095R+1G-0.762B optimally separated plant cells and *B. lactucae* and was used in further image analysis. This optimum *B. lactucae* contrast image was then 3D Gaussian filtered to enhance the signal of the dense hyphae compared to that of thin plant cell outlines. Non-*B. lactucae* leaf veins and stomata that appeared bright in the dOCT image due to their presumed higher sub-cellular activity were segmented out manually. After performing a global intensity threshold, small objects were removed with an automatic filter resulting in a clean segmentation of the large majority of *B. lactucae* hyphae. The *B. lactucae* voxels were summed to obtain their volume, while the number of voxels after skeletonization was summed as a measure of the total hyphae length.

Quantitative phenotyping was performed by investigating the presence and calculating the volume, and length of the *B. lactucae* hyphae in each of the 48 3D dOCT images, see Fig. 2c. Variety Bedford had 4/16 volumes colonized. Within these few infected Bedford samples, the degree of infection was low, with a volume below 0.2 nl/mm$^2$ and length below 3 mm/mm$^2$. This indicates that Bedford resists infection thereby limiting *B. lactucae* hyphal growth. Variety Iceberg had 6/16 volumes colonized, 50% more than Bedford. Also *B. lactucae* hyphal colonization was slightly increased as shown by the larger hyphae volume up to 0.6 nl/mm$^2$ and length up to 8 mm/mm$^2$ (and in Supplementary Figs. 6 and 7), but the difference was not statistically significant ($p = 0.384$ for both volume and length).

In contrast, variety Salinas had 15/16 colonized volumes, and the *B. lactucae* hyphae were present throughout the whole imaged volumes. This is also reflected in the quantified volume reaching up to 2.3 nL/mm$^2$ and length reaching up to 18 mm/mm$^2$, which is significantly different from that in both Iceberg ($p = 3.83 \times 10^{-4}$ for volume and length) and Bedford ($p = 1.42 \times 10^{-4}$ for volume and length).

We compared our hyphal volume and length infection level quantification to the qPCR analysis, which is commonly used to quantify the presence of disease tissue in plant tissue[19,24] We applied qPCR analysis on *B. lactucae*-infected leaves from the same three cultivars (5 leaves per cultivar) in an independent experiment. The measured level of pathogen infection, Fig. 2d, showed the same pattern in relative magnitude as the hyphal volume and length, with Iceberg being slightly more infected than Bedford ($p = 0.104$) and Salinas showing high infection levels ($p = 2.90 \times 10^{-2}$ compared with either genotype). An important distinction between the two methods is that the qPCR analysis represents an average quantification across a whole leaf, while the areas imaged with dOCT are a small fraction of a leaf disc. As a consequence, dOCT images reveal the heterogenous distribution of B. lactucae hyphae within the tissue, which is reflected by the presence of a number of uncolonized image areas mainly for Bedford and Iceberg. The comparison with qPCR data shows that dOCT-based quantification of hyphal volume and length is in agreement with the pathogen-plant DNA ratio.

In contrast to the destructive qPCR analysis, dOCT also gives immediate access to the underlying plant and pathogen structure in live tissue in 3D. Even though the difference in quantitative disease severity between Bedford and Iceberg is not significant, qualitative investigation of the imaged volumes could help to identify potential differences in the resistance responses. For example, around the hyphae in the Bedford samples dead plant cells were commonly observed, see Fig. 2e, pointing to hypersensitive cell death as part of the resistance response[22,23]. Iceberg, in contrast, had only living plant cells around hyphae, pointing to a different resistance mechanism. We did occasionally observe hyphae shaped structures with a similar spectral composition as dead plant cells in infected Iceberg samples (annotated as inactive hypha). Although we have not further investigated these structures they may be part of an Iceberg-specific response to *B. lactucae* infection.

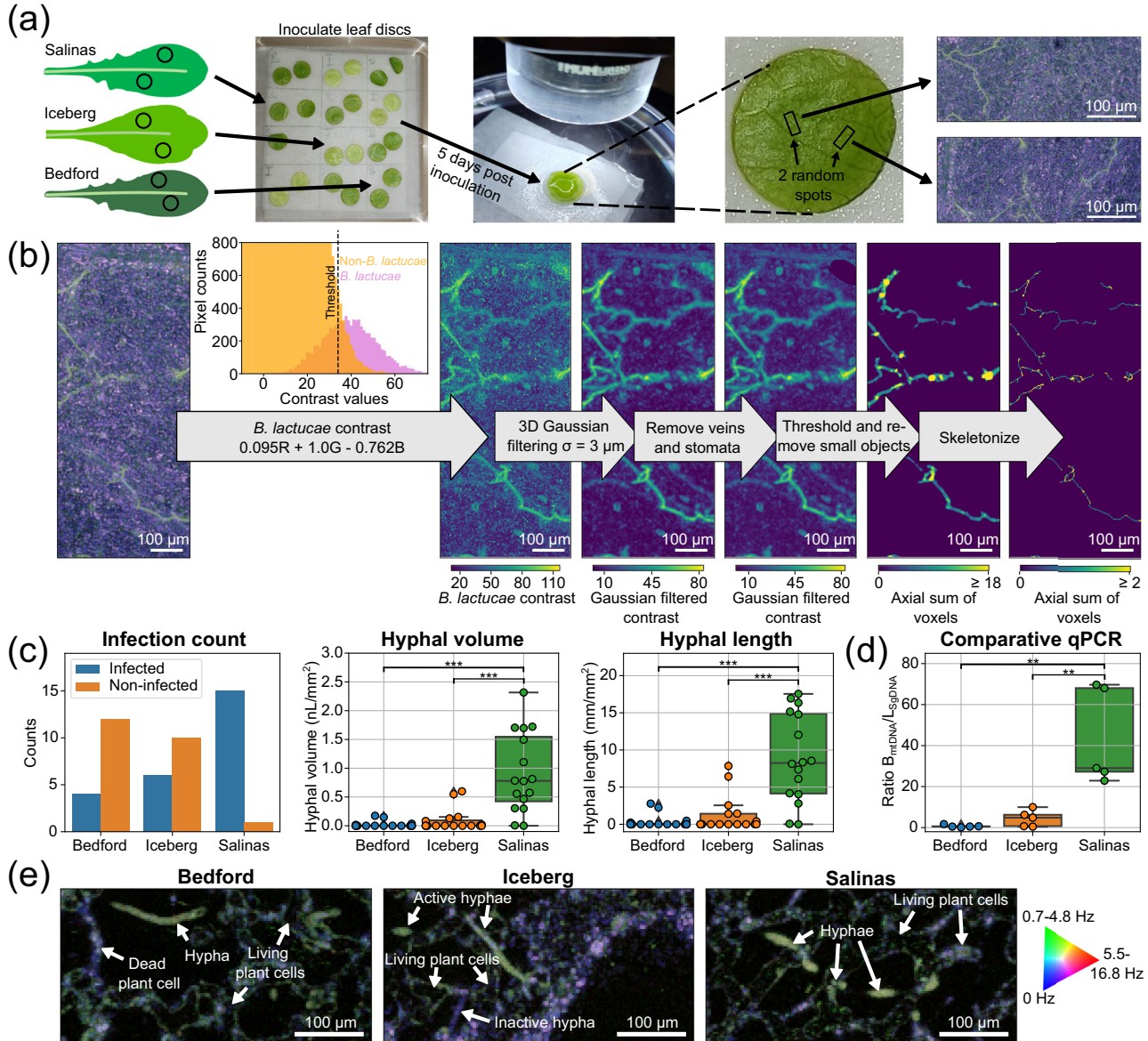

**Fig. 2 | Micro-phenotyping with dynamic OCT. a** Overview of the dynamic OCT (dOCT) imaging process for digital phenotyping. **b** *Bremia lactucae* segmentation pipeline for dynamic OCT volumes. The images are a maximum intensity projection (before threshold) or an axial sum of the non-zero pixels (after threshold). Images for all imaged segments after 3D Gaussian filtering and after threshold and removing small objects are shown in Supplement Figs. 6 and 7. Supplementary movie 2 shows a 3D rendering of the volume data after each segmentation step. **c** dOCT-based degree of *B. lactucae* infection quantification per volume for each genotype ($N = 16$ per genotype, statistical analysis was performed using a pair-wise two-sided Mann–Whitney U test ***$p < 0.001$). **d** qPCR data for relative pathogen DNA content for the three studied genotypes ($N = 5$ per genotype, statistical analysis was performed using a two-sided Welch's t-test assuming unequal variance *$p < 0.05$). For both (**c**) and (**d**), P-value correction for multiple testing was performed using the Benjamini−Hochberg method. See Supplementary Table 2 for test statistic values. For the box plots, the middle line represents the median value, the lower and upper quartile lines represent the 25th and 75th percentile, and the whiskers show the maximum and minimum values of the data that are not qualified as outliers. Outliers are visible as points outside the whiskers. Source data for (**c,d**) are provided in a Source Data file. **e** Three typical horizontal dOCT cross sections with hyphae for each genotype.

Importantly, dOCT could visualize and quantitatively assess the level of infection in vivo before the onset of visibly observable symptoms. This precise and direct quantification of pathogen proliferation has been conventionally restricted to destructive methods, such as microscopic analysis of stained tissue or DNA-based quantification using qPCR.

### Longitudinal imaging of pathogen growth

Finally, we demonstrate quantitative assessment of temporal development of *B. lactucae* by in vivo 3D imaging of infected leaf disks from the susceptible Salinas variety. Multiple measurements of the same area, taken over the course of two to three days starting from the end

of the third day post inoculation, visualize the development of the 3D *B. lactucae* hyphal network inside lettuce leaf tissue, see Fig. 3a.

The infection growth pattern is summarized in a 3D overview image, Fig. 3b, showing hyphae segmented from the dOCT images with the color indicating the growth time. Within a single time interval of 12 h often long hyphae of hundreds of $\mu$m are formed, which then either stop growing or grow into a deeper layer out of the depth of field. Additionally, the volume over time of *B. lactucae* hyphae in three leaf segments shows rapid colonization that reaches a maximum around day 6. It demonstrates hyphal growth quantification allowing one to follow the progression of the plant's colonization by *B. lactucae* in time, Fig. 3c and Supplementary Fig 9.

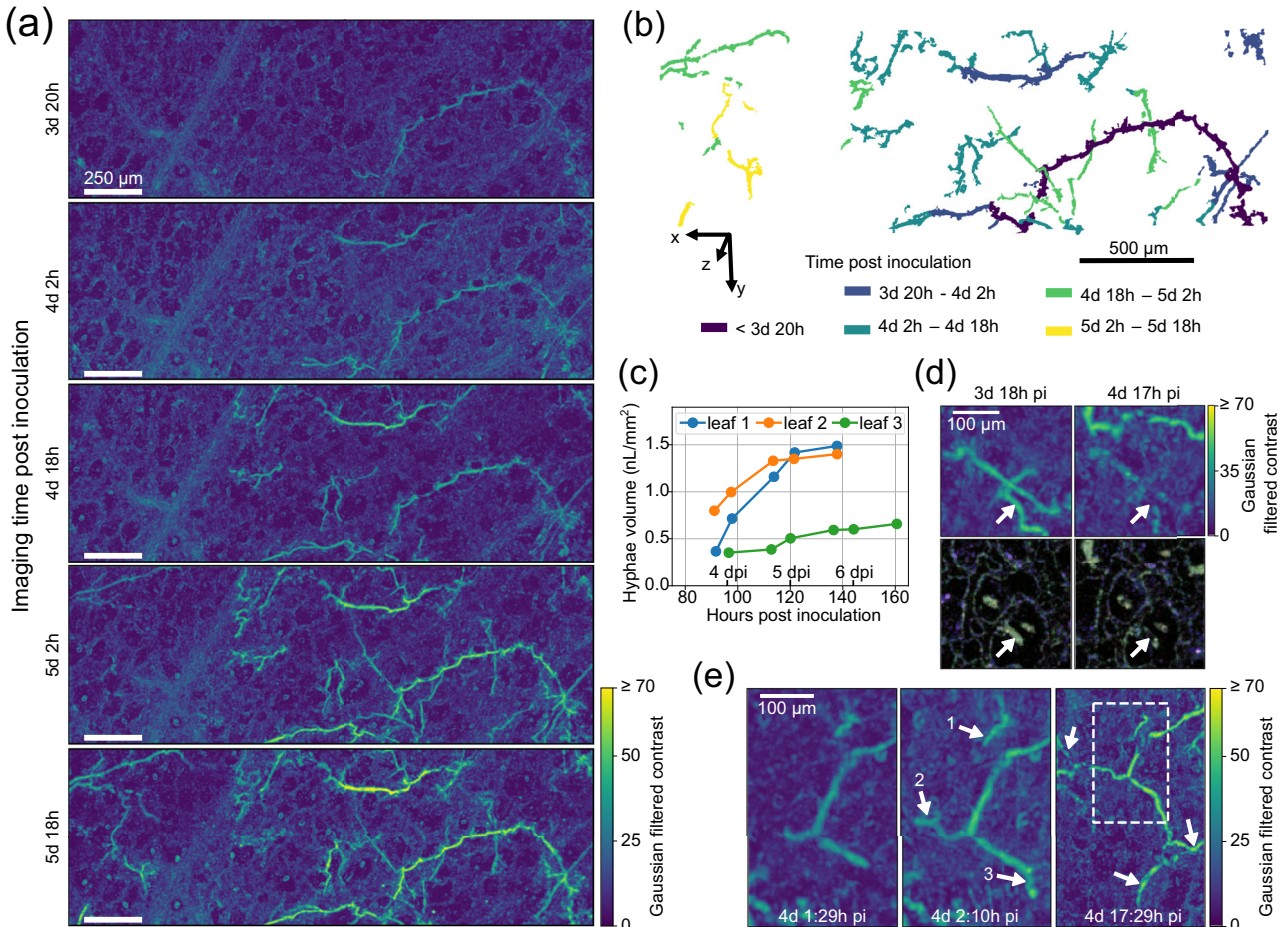

**Fig. 3 | Longitudinal imaging of pathogen growth. a** The progression of *Bremia lactucae* infection over two days visualized by the maximum intensity projection (MIP) of Gaussian filtered *B. lactucae* contrast images. **b** The segmented point cloud of *B. lactucae* hyphae, colored by the time period of growth. **c** Total *B. lactucae* hyphae volume increase over time for three different leaf segments (one shown in (**a**), the others in Supplement Figure 9). **d** A close-up of hyphae that lost their

activity (white arrow) while new active hyphae develop. At the top, the MIP of Gaussian filtered *B. lactucae* contrast and at the bottom the horizontal cross-sectional dOCT images at the hyphae depth. **e** A close-up of two images taken with 40 minutes in between, and a zoom-out image 15 hours later. The arrows indicate the newly grown hyphae and the white-dashed box the area of the first two images. Source data for (**c**) are provided in a Source Data file.

Longitudinal *B. lactucae* imaging also revealed hyphae that lost their medium frequency dOCT signal almost completely over the course of a day. Figure 3d shows a hypha that first displayed a strong medium frequency signal (green) throughout its volume, which is lost 23 hours later when only a weak signal remains close to the hyphal wall. The newly grown bright hypha (top of image) shows that the loss of activity is not an imaging artifact. We observed this effect also in other time-lapse images, where, especially after sporulation, many hyphae lost dOCT signal. These inactive hyphae might be left devoid when the cytoplasm within the coenocytic hyphae is withdrawn and moved towards the sporangiophores[25]. Although this can complicate the precise quantification of all hyphae, it can also provide unique insights into the physiology and development of *B. lactucae* during different phases of infection.

Longitudinal *B. lactucae* growth can also be detected at short time scales, Fig. 3e. In 40 minutes, three hyphal tips (1–3) grew at an average speed of 54 $\mu$m/h, 105 $\mu$m/h, and 115 $\mu$m/h, respectively. The same area visualized 15 h later shows that the hyphae of tips 2 and 3 ($\approx +0.5$ mm in 15 h) have continued to grow and branched out (arrows), while the hypha at tip 1 barely grew further.

These examples illustrate that dOCT enables capturing spatial and temporal dynamics of plant-pathogen interactions at a high resolution at different time scales. Moreover, the 3D analysis allows to identify individual hyphae, their interactions, and their branching behavior.

This enables studying pathogen growth mechanisms in vivo at a microscopic level, e.g., in relation to plant structures or plant immune responses, which could yield new insights into pathogen-plant interactions.

## Discussion

Dynamic OCT can image plants and pathogens in vivo in 3D, with high contrast, at cellular resolution, and without requiring any labeling. This enabled quantifying the spread of downy mildew infection, and following the progression of the infection in the same plant tissue over time.

Imaging the pathogen rather than secondary symptoms gives a high accuracy in quantifying disease severity. Two factors that affect the precision of our method are the spatial resolution and the segmentation accuracy. Due to the high spatial resolution of 3 $\mu$m of our OCT imaging system, we could image thin hyphae with diameters down to 10–20 $\mu$m with sufficient accuracy. OCT systems with a poorer lateral and axial resolution will introduce a significant inaccuracy into the results or would even not be able to visualize small hyphae. An important factor in the segmentation accuracy is the choice of threshold. The threshold used to separate plant and pathogen tissue removes up to 35% of the pathogen hyphae (see Supplementary Fig. 3), thus underestimating the actual hyphal volumes. Therefore, it was essential that all the samples within an experiment were processed

with the same reference values and threshold, such that a possible bias would affect all cultivars in the same way. The decrease of hyphae volume and length with increasing global dOCT signal threshold, shown in Supplementary Section 2 and Supplementary Fig. 5, was similar for all three genotypes, as seen in constant p-values with varying thresholds. Thus the comparison between cultivars and samples remains justified. We anticipate that the rapid development of deep learning and machine learning tools will further aid and speed up the analysis and interpretation of images by improved accuracy of segmentation and classification[26].

The accurate quantification of disease severity in the imaged dOCT volume could be used for micro-phenotyping to enhance precision in disease resistance screens. We showed the potential of dOCT for micro-phenotyping by measuring downy mildew colonization levels in three lettuce varieties and confirming the results with qPCR-based measurements of pathogen DNA. Increasing the number of imaged samples will likely enable resolving also minor differences in resistance level. Moreover, the functional and structural volumetric data obtained with dOCT can aid the discovery and interpretation of resistance mechanisms.

For longitudinal studies on longer time scales, dynamic OCT could also be applied to attached leaves of (small) plants or seedlings when the leaf is stabilized to avoid parasitic dynamic signal. Infiltration with water or PFD, essential for obtaining high-quality images of leaves with gas-exchange cavities[15], will be more challenging, and immersion of the leaf surface requires a horizontal interface and/or a special mount. Imaging stems or roots on a substrate would be easier as, in the absence of gas-exchange cavities, there is no need for infiltration. Here, we chose to image leaf disks because *B.lactucae* primarily infects leaf tissue. Moreover, because leaf disks allow for controlled infection and are convenient to handle they are commonly used in *B. lactucae* resistance tests[27].

In conclusion, dynamic OCT pathogen imaging opens up the opportunity for micro-phenotyping at cellular, tissue, and organ level in relation to plant disease susceptibility and resistance, e.g., by quantifying colony size, haustoria development, and disease morphology indices[28]. Its real-time and in vivo imaging capability also facilitates the study of pathogen responses to local immune reactions or interventions such as the administration of pesticides or resistance-stimulating microbes. Measuring and understanding the localized dynamics of pathogen infection in this way can assist the development of successful resistance breeding strategies. Besides structural imaging of pathogens and plants, dynamic OCT gives local intrinsic functional contrast on the activity of the plant cells and pathogens. This may enable a more detailed analysis of biological and molecular processes inside living plant and pathogen cells during the infection process.

## Methods
### Sample preparation and experimental design
Lettuce plants were cultivated in pots with potting soil at 21 °C under long-day conditions (16 h light, 100 $\mu$mol/m$^2$/s, 70% humidity). Leaf disks (10 mm diameter) from four-week-old plants were placed, abaxial side up, in petri dishes on moist filter paper and inoculated with *Bremia lactucae* (race Bl:33EU[29], obtained from Rijk Zwaan B.V., De Lier, The Netherlands) by spraying with spore suspension (40 spores/$\mu$L, experiment 1) or by dabbing with *B. lactucae*-infected lettuce cotyledons that were covered with sporangiophores (experiment 2). Inoculated disks were incubated at 16 °C under short-day conditions (9 h light, 100 $\mu$mol/m$^2$/s). For experiment 1, leaf disks from three cultivars (Bedford, Iceberg, Salinas; seeds obtained from Rijk Zwaan B.V., De Lier, The Netherlands) were sampled at 5 dpi, vacuum infiltrated with tap water, and imaged with a water droplet on the surface to reduce surface reflection. In experiment 2, disks of cultivar Salinas were imaged twice daily, at the beginning and end of the light period,

between 4 and 7 dpi. For imaging, disks were submerged in perfluorodecalin (PFD, Sigma–Aldrich) in a custom mount, and covered with a cover slip. PFD infiltrates the air spaces in plant leaves without applying a vacuum because of its low surface tension[21]. After imaging, disks were rinsed with tap water and returned to the growth chamber. PFD infiltration had no visible effects and disappeared within 30 min. To visualize *B. lactucae* colonization with brightfield microscopy, leaf tissue was stained with trypan blue[30]. Germinating *B. lactucae* spores for imaging (Fig. 1c) were prepared by dabbing lettuce seedlings covered in sporangiophores onto water agarose pads (1.5% agarose) to deposit spores and incubation at 16 °C in the dark for 2 h. During OCT imaging, the spores were submersed in water to reduce deformation and surface reflection intensity. For differential interference contrast microscopy the agarose pads with germinating spores were covered with a cover slip and imaged with an inverted wide-field microscope (Eclipse Ti, Nikon) equipped with a 40× oil immersion objective (Plan Fluor 40x/1.30 Oil DIC, Nikon).

*Arabidopsis thaliana* Col-0 plants were grown in pots with potting soil at 21 °C under long-day conditions (16 h light, 100 $\mu$mol/m$^2$/s, 70% humidity). Plants at the age of 14 days were inoculated with *Hyaloperonospora arabidopsidis* isolate Noco2[31] by spraying with spore suspension (40 spores/$\mu$L) and incubated in trays with clear lids at 16 °C under short-day conditions (9 h light, 100 $\mu$mol/m$^2$/s) until imaging at 5 dpi. A single true leaf was detached, submerged in PFD as described above, and imaged. For imaging the root-knot nematodes inside pepper roots, pepper plants (*Capsicum annuum*, seeds from Rijk Zwaan B.V.) were sown in a sandy soil mixture in a greenhouse that was set at 25 °C and an 18-hour lighting regime (Son-T). After three weeks, the plants were transplanted in a sandy-soil mixture that was infested with nematode eggs (*Meloidogyne incognita*, approximately 5.000 eggs per plant) in a greenhouse set at 23 °C and an 18-hour lighting regime (Son-T). Six weeks later, roots were washed, kept in water and imaged within a day from uprooting. An infected gall was chosen based on inspection with the naked eye, and imaged while submersed in a small layer of water to reduce the optical reflections at the outer surface. The radish plants (*Raphanus raphanistrum subsp. sativus*) were grown and infected at Rijk Zwaan B.V., De Lier, The Netherlands. Two-week-old plants were inoculated with *Hyaloperonospora brassicae f. sp. raphani* by spraying a suspension of spores (50 spores/$\mu$L) that were washed off from radish leaves from a previous infection cycle. After infection, the plants were kept in a growth chamber at 15 °C on a 14 h light (100 $\mu$mol/m$^2$/s) per day regime and covered with a transparent plastic sheet to maintain high humidity. At 6 days after inoculation, the cotyledon leaves were infiltrated with tap water using a syringe[15] and imaged.

### Infection quantification with qPCR
To quantify *B. lactucae* infection level in lettuce with a Taqman qPCR assay as alternative method, the youngest fully developed leaf of 3.5-week-old plants (usually the third true leaf) was spray-inoculated with sporangia suspension (50 sporangia/$\mu$L, race Bl:33EU), covered with transparent plastic lids, and moved to a short day condition growth chamber (16 °C, 9 h tube light, 100 $\mu$mol/m$^2$/s). To determine relative pathogen DNA content, the inoculated leaves were sampled at 7 dpi, ground to a fine powder in liquid nitrogen with mortar and pestle, and a 40-50 mg subsample of the resulting powder was used for DNA extraction. DNA was extracted with the MagMAX Plant DNA isolation kit (Thermo Scientific) and a KingFisher Flex Magnetic Particle Processor with 96 Deep-Well Head (Thermo Scientific) following the manufacturer's instructions. Polyvinylpyrrolidone was added to "Lysis Buffer A" to a 2% (w/v) final concentration as recommended for polyphenol-rich samples. As input for a TaqMan qPCR assay the eluate was diluted 10-fold with Milli-Q water to obtain DNA concentrations between 0.5 and 5 ng/$\mu$L. The *B. lactucae* primer pair, Blorf286-F (5′-AGCTAGATTTACCACATACG-3′) and Blorf286-R (5′-CAAGAGGAGTCAT

TGTTTG-3′), and internal probe, Blorf286-INT (5′-CAGTGGGA-CATTCTATTGATGAAGA-3′) target a mitochondrial gene (NCBI Gene ID: 38665116) and were modified from Kunjeti et al.[32]. The lettuce primer pair, LsTUB1-F (5′-TGAAACTCTGTGATGTCAAC-3′) and LsTUB1-R (5′-TGAGAGACCTACATACACCA-3′), and internal probe, LsTUB1-INT (5′-AACGTAGGGAGGTTGTTAAAGATGA-3′) target a lettuce genomic alpha-tubulin gene (NCBI Gene ID: 111903327, reference assembly Lsat_Salinas_v7). The probes were labeled with FAM-6 (lettuce) or HEX (*B. lactucae*) fluorescence dyes on the 5′-end and double-quenched with the internal ZEN quencher and at the 3′-end with the Iowa Black quencher (PrimeTime qPCR probes, Integrated DNA Technology). The 5 µL reaction volumes contained 2.5 µL polymerase master mix (PrimeTime Gene Expression Master Mix, Integrated DNA Technology), 0.15 µL of 10 µM lettuce primer solution, 0.075 µL of 10 µM *B. lactucae* primer solution, 0.075 µL of each 10 µM probe, 0.9 µL nuclease-free water and 1 µL DNA template. All reactions were run in duplicate on a single 384-well plate (4309849, Applied Biosystems). The qPCR run was conducted in the ViiA 7 Real-Time PCR System (Applied Biosystems) with the PCR cycle conditions 50 °C for 2 min, 95 °C for 3 min, 40 cycles of 95 °C for 15 s and 60 °C for 1 min. Using the mean cycle threshold values of the duplicate reactions we calculated the ratio of *B. lactucae* mitochondrial DNA to lettuce genomic DNA as $2^{(Ct\ LsTUB1\ -\ Ct\ Blorf286)}$.

## Imaging setup and dOCT processing

The OCT measurements were done with a Ganymede II HR spectral domain OCT setup (Thorlabs, Germany), also used in previous work[15], with a 36 kHz A-scan rate and a center wavelength of 900 nm and a 160-nm bandwidth, giving a 2.1 µm axial resolution in tissue. The OCT scan lens (OCT-LK2-BB, Thorlabs, Germany) that was used is designed for this setup and has a high lateral resolution of about 3 µm. B-scan data were acquired using ThorImage (version 5.4.1, Thorlabs), and 3D dynamic OCT data was acquired using a Python wrapper around a C++ module that was included in the ThorImage package. 3D scans were automatically divided into blocks of around 40 dynamic B-scans that were saved before starting the next block.

For experiment 1 (Fig. 2), the dynamic B-scans used 100 raw B-scans with 14.6 ms repetition rate (1.46 s per dynamic B-scan). The B-scans contained 360 A-scans over 1.08 mm length, and 150 dynamic B-scans were obtained over 0.45 mm width (3 µm lateral sampling in both directions). The volume acquisition time was 4.9 min. For experiment 2 (Fig. 3), the B-scan repetition time was increased to 29.3 ms, and only 50 raw B-scans were obtained in the same 1.46 s. The increased repetition time was possible because the signal at frequencies above 16.4 Hz was not used for the dOCT contrast. The B-scans contained 887 A-scans over 2.66 mm length, and 300 dynamic B-scans were obtained over 0.90 mm width, giving a volume acquisition time of 7.3 min. Saving the raw data in between the blocks took another few minutes per volume. The nematode-infected root and infected radish leaf were imaged in the same way as experiment 1 (100 B-scans with 14.6 ms repetition rate and 3 µm resolution), and consisted of respectively 100 (0.3 mm) and 200 (0.6 mm) dynamic B-scans of 1.08 mm in length direction. The infected Arabidopsis leaf was imaged with the same settings as experiment 2.

The raw OCT data per B-scan was first processed in the conventional way, consisting of subtracting the reference spectrum, linearization in the wavenumber domain, spectral reshaping with a Hanning window, dispersion correction, an inverse discrete Fourier transform (DFT), and taking the absolute value. For the 50 or 100 B-scans at the same location, a DFT was taken in the time direction and the amplitude spectrum (AS) was obtained by taking the absolute value. The log of the average value of the AS in the three frequency bands of 0 Hz (blue), 0.7–4.8 Hz (green), and 5.5–16.4 Hz (red) was taken as intensity for the color channels. Note that the 0 Hz signal is equal to the average amplitude of the OCT signal over time. The green and red channels

encode fluctuations at different frequencies. Log compression requires a top and bottom limit, which were determined for each color channel separately, as described in Supplementary Section 1.2 and Supplementary Fig. 2. In short, the top limit was the mean of the highest value over all volumes within the experiment (excluding coverslip interfaces), while the lower limit was the average noise level determined as the peak location of the image amplitude histogram before log compressing. The chosen limits gave a dynamic range of 61.6 dB, 50.7 dB, and 41.2 dB for the low, medium, and high frequencies respectively, while for the time lapse images the dynamic ranges were 63.0 dB, 52.0 dB, and 45.2 dB. For display purposes, the contrast of the dOCT images in the figures is enhanced by saturating the higher values above 180 or 200 (out of 255).

## Image segmentation

The segmentation for experiment 1, the quantification of infection, was done in Python 3.7 according to the pipeline in Fig. 2b. The sigma of the Gaussian filter kernel was chosen to be 1 pixel in the lateral direction and 3/1.37 pixels in the axial direction, corresponding to 3 µm isotropically.

Masks to filter out veins, stomata, non-pathogen active cells and a few artifacts were obtained after the threshold was determined (see Supplementary section 1.3 and Supplementary Fig. 3). After thresholding and removing small objects, the non-pathogen structures that were left were removed manually from the axial sum image in imageJ by an expert. When structures were not clearly stomata or veins without infection, the expert carefully investigated in the 3D dOCT image stack whether it could be a *B. lactucae* hyphae or plant cell. This was determined based on the shape of the structure and the connection with surrounding plant cells and, if present, surrounding *B. lactucae* hyphae. The comparison between Supplementary Figs. 4,6,7 shows the removed structures. The axial sum image was then translated to a mask that was used after the Gaussian filtering in the segmentation pipeline.

The threshold was chosen at 34 (based on color values ranging from 0-255 and top and bottom reference values as discussed above). Objects smaller than 256 voxels (with a connectivity of 4 voxels) were removed with an automatic morphology filter. The volume was then obtained by counting the number of remaining voxels, multiplying this number with the physical voxel volume, and dividing it through the imaged surface area to get the volume per imaged area. The length was obtained by skeletonizing the segmented hyphae and multiplying it with the lateral pixel width of 3 µm. This is an approximation as on a diagonal, the length could be $\sqrt{2}$ larger, and vertically the distance between pixels is more than a factor 2 lower. The potential error is probably limited as most hyphae grow mostly in the horizontal direction. Moreover, a potential bias would be similar for different genotypes making the relative comparison of colonization still accurate.

For the time lapses, the step of removing veins and stomata was first omitted, and we chose a threshold of 22, which could capture most hyphae. The threshold is lower than for the quantitative data because of the slightly different dynamic range and (especially for the blue channel) a slightly lower noise reference value. Moreover, veins and stomata could be filtered out in 3D from the point cloud, allowing for slightly more noise that could be segmented out. The raw point clouds were further processed using CloudCompare (V2.13.alpha). The point cloud of the last image of a time series was manually cleaned up by removing stomata, veins, and noise that was no *B. lactucae*. Hyphae that had become inactive or hyphae that were more clearly imaged in earlier images were added and aligned to obtain a point cloud that was as complete as possible. Then the point cloud was divided based on the time period that the hyphae had grown, see Fig. 3b. For cleaning up the point cloud, adding hyphae, and determining the time of growth, the 3D dOCT image stack was

used as a guide to check for the presence of hyphae. This segmented point cloud is then used to quantify the *B. lactucae* colonization over time, by multiplying the amount of points in the point cloud with the voxel volume and dividing through the total imaged leaf area. The growth speed between the images in Fig. 3e was estimated manually based on the dOCT-contrast image stack in ImageJ.

## Statistics

3D image stacks with dOCT contrast were manually inspected by an expert to validate the presence or absence of infection, Fig. 2c, left. The number of samples, means, standard deviations, minimum/maximum of the hyphal volume, hyphal length, and qPCR ratio are shown in Supplementary Table 1. Differences in hyphal volumes and lengths between host genotypes were evaluated with a two-sided Mann–Whitney U test, which is non-parametric and does not assume normally distributed data, and for which all samples were included ($N = 16$ per genotype for dynamic OCT), including those without infection. For qPCR data, differences between host genotypes were evaluated using two-sided Welch's t-tests assuming unequal variance ($N = 5$ per genotype). *P*-value correction for multiple testing was performed using the Benjamini–Hochberg method. Test statistic values, *p*-values, effect sizes, and effect size 95% confidence intervals are shown in Supplementary Table 2. Statistical test values and test statistics were obtained with R4.4.0.

## Reporting summary

Further information on research design is available in the Nature Portfolio Reporting Summary linked to this article.

## Data availability

Additional analysis and visualizations are available in the Supplementary Materials. A reporting summary of the article is available as a supplementary file. A representative selection of the data, all the dynamic OCT volumes, and supporting code for data processing and plotting have been uploaded to a freely-accessible Zenodo repository[33]. [https://doi.org/10.5281/zenodo.11428245]. Source data are provided with this paper.

## Code availability

Code for dynamic OCT processing, segmentation, and data analysis, together with a representative selection of the data is available in a Zenodo repository[33]. [https://doi.org/10.5281/zenodo.11428245].

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

## Acknowledgements

This work was funded by Partnership programme Dutch Research Council (NWO) Domain Applied and Engineering Sciences - Rijk Zwaan B.V. (16293, J.K. and G.v.d.A.). We thank Rijk Zwaan B.V. for providing seeds, infected plants and useful discussions. We thank Andrew Pape for his help in developing the qPCR screening method. We thank Ron Hoogerheide for technical support.

## Author contributions

J.d.W. designed and implemented the dynamic OCT imaging method and contrast optimization. J.d.W., S.T. and M.R.S. carried out the experiments and did the analysis. J.K. and G.v.d.A. acquired funding and supervised the project. All authors (with J.d.W., S.T. and J.K. in the lead) co-wrote the manuscript and contributed to the revision of the manuscript.

## Competing interests

The authors declare no competing interests.
