## [Peer Review File · Nature Communications]

REVIEWER COMMENTS

Reviewer #1 (Remarks to the Author):

This work by Joe De Wit et al. introduces a novel approach to investigating in vivo pathogen dynamics in plants by using dynamic optical coherence tomography (dOCT). The manuscript is well written and structured, effectively demonstrating the capabilities of dOCT in real-time, 3D imaging of plant leaves and the pathogen dynamics. While the results are presented clearly, and the methods are detailed, I have several comments to address before considering this paper for publication.

1. The authors optimized the chosen weights (0.3R+1.3G-0.8B) empirically for separating plant cells and *B. Lactucae*. It would be beneficial to provide additional details on the criteria used in this optimization process to enhance the clarity of the methodology.
2. On page 3, the authors mentioned that “Non-*B. lactucae* leaf veins and stomata that appeared bright in the dOCT image due to their presumed higher sub-cellular activity were segmented out”. A more comprehensive description of how this segmentation was achieved is recommended. Furthermore, details about the global intensity threshold and whether it is consistent across different samples should be clarified.
3. The calculation of Hyphae volume involves counting *B. lactucae* voxels and summing them to obtain the volume. Given the dependency on the global intensity threshold (as raised in the previous question), the authors are encouraged to conduct an analysis of the errors associated with this calculation. In addition, considering the OCT system’s lateral resolution ($\sim 3 \mu\text{m}$) is comparable to the Hyphae diameter, leading to potential inaccuracies, it would be valuable for the authors to discuss potential errors in the Hyphae volume measurement.
4. Considering the limited number of samples in this study, a thorough statistical analysis is crucial to establish the significance of the results. Therefore, it is recommended that the authors provide a detailed statistical analysis for the results presented in Figure 2 to strengthen the robustness of their findings.

Minor comments:

1. In Fig. 2(b), it should be 0.3R+1.3G-0.8B inside the arrow, to be consistent with the context.
2. It is recommended to include color bars for all the intensity maps in Figs. 2 and 3.

Reviewer #2 (Remarks to the Author):

Manuscript NCOMMS-23-64139 follows primarily the obligate pathogen *B. lactucae* in plant tissue without staining or fluorescent protein tags. This work demonstrates the potential utility in clarifying

the disease cycles of these types of pathogens and in quantifying the level of resistance between cultivars. The methodology employed enabling 3D in vivo pathogen dynamics at increased resolution has potential application in visualizing molecular processes that occur during pathogen-host interactions.

Page 1, 2nd paragraph of introduction: The authors mention that measuring *B. lactucae* “enabled precise disease severity quantification”. It is difficult to show precision when the cultivars used are either resistant or susceptible with no in between. The results in Fig 2C show increase hyphal volume in cultivar Salinas, and significantly less so in both of the resistant cultivars. The gradation of level of resistance between the resistant cultivars was not resolved. This sentence should be rephrased to indicate correlation between hyphal volume of *B. lactucae* and resistance or susceptibility.

Page 7, Methods: Is there a reference for *B. lactucae* race Bl:33EU? Please insert. Also, what was the source of Bedford, Iceberg and Salinas from Figure 2? Couldn't find this information in the supplemental file either.

Page 3: What is “good contrast”? Perhaps contrast alone is a better choice.

Page 5, Figure 3: The authors indicate over 2 days in (a) but the labeling indicates 5d 18h, 4d 18h etc. Please correct, rewrite, or explain.

Page 5, Discussion: The phrase in the first sentence “something that has not been shown before” is not necessary.

Page 5, Discussion: Agree that a major benefit of the approach can be following infection in the same plant tissue over time. Would there be a way to adapt the approach be applied to attached leaves of a whole plant rather than detached leaf discs? Also, did the authors consider testing other lettuce tissue besides leaves, to determine if the colonization is systemic?

Page 6, Methods: At bottom of page: what is a “good amount”? Perhaps sufficient amount is a better word choice.

Reviewer #3 (Remarks to the Author):

Dear authors,

The article is well-written, illustrating the potential application of OCT in capturing the spatial and temporal dynamics of host-pathogen interactions, as seen with the fungus *Bremia lactucae*. The videos are quite illustrative.

However, peculiarly, and lacking essential methodological details, the authors present findings involving downy mildew (*Hyaloperonospora arabidopsidis*) in *Arabidopsis* leaves and the root-knot nematode *Meloidogyne incognita* in pepper roots (crucial information such as acquisition

methods, specific varieties involved, pathogen and parasitic nematode density, etc., is notably absent). Even new pathogen-host combinations are also added to the supplementary material, which doesn't make sense without a prior introductory note and without methodological details. To address this, either these segments should be omitted from the results and discussion, or more information need to be incorporated, including pertinent details, into the materials and methods section. Furthermore, a more in-depth analysis of the obtained results with all the pathogens/ plant parasitic nematodes is warranted. I would like to draw attention to the fact that, being a plant pathologist, I do not possess expertise in the more technical aspects. However, regarding the pathogen/parasite-host interaction, I couldn't help notice this omission in the manuscript. Please, consider my suggestions throughout the text and highlight the necessary changes in the revised version. Also, it should be noted that the figures must be self-standing, and in this regard, all names should be written in full, with acronyms previously detailed.

I wish you good work,

We thank the referees for carefully reading our manuscript. Below we reply on a point-by-point basis to the remarks made by all the reviewers.

Reviewer #1 (Remarks to the Author):

This work by Jos De Wit et al. introduces a novel approach to investigating in vivo pathogen dynamics in plants by using dynamic optical coherence tomography (dOCT). The manuscript is well written and structured, effectively demonstrating the capabilities of dOCT in real-time, 3D imaging of plant leaves and the pathogen dynamics. While the results are presented clearly, and the methods are detailed, I have several comments to address before considering this paper for publication.

1. The authors optimized the chosen weights (0.3R+1.3G-0.8B) empirically for separating plant cells and B. Lactucae. It would be beneficial to provide additional details on the criteria used in this optimization process to enhance the clarity of the methodology.

We agree with the reviewer and have thoroughly analyzed the optimization method. The whole optimization process is explained in a flow chart in Supplementary Fig. 3, complemented with supporting text in Supplementary section 1.3. Moreover, we have clarified the choice of the dynamic range lower and upper dOCT signal limits in more detail in Supplementary section 1.2 and Supplementary Figure 2. The analysis of the optimization method resulted in minimal changes to used threshold and the segmented volume and length.

2. On page 3, the authors mentioned that “Non-B. lactucae leaf veins and stomata that appeared bright in the dOCT image due to their presumed higher sub-cellular activity were segmented out”. A more comprehensive description of how this segmentation was achieved is recommended. Furthermore, details about the global intensity threshold and whether it is consistent across different samples should be clarified.

We agree. We have added a more detailed description of the segmentation method to the manuscript methods section, including an elaboration on the step of manually removing veins, stomata and artefacts. The method of choosing a global threshold is elaborated on in Supplementary section 1.3 and Supplementary Figure 3.

3. The calculation of Hyphae volume involves counting B. lactucae voxels and summing them to obtain the volume. Given the dependency on the global intensity threshold (as raised in the previous question), the authors are encouraged to conduct an analysis of the errors associated with this calculation. In addition, considering the OCT system’s lateral resolution (~3 μm) is comparable to the Hyphae diameter, leading to potential inaccuracies, it would be valuable for the authors to discuss potential errors in the Hyphae volume measurement.

We have added a paragraph in the discussion section of the manuscript, discussing the accuracy of our method (lines 158-168). Moreover, we added an analysis of the dependency of hyphae volume and length on the threshold in Supplementary section 2 and Supplementary Figure 5.

4. Considering the limited number of samples in this study, a thorough statistical analysis is crucial to establish the significance of the results. Therefore, it is recommended that the authors provide a detailed statistical analysis for the results presented in Figure 2 to strengthen the robustness of their findings.

We extended the statistical analysis by providing test statistics, applying a P-value correction for multiple testing, and providing metrics of the data per genotype. We added a section in the methods describing the used statistical methods, details on the statistical tests in the legend of Figure 2 and added Supplementary Tables 1 and 2 providing the statistics of the data per genotype and of the test results.

We also added qPCR data from an independent experiment, Fig. 2(d), of which the results are in line with our findings and show a similar variation as we found with quantifying the hyphal volume and length. It ranks the genotypes in the same order for disease presence. Moreover, we added data highlighting some qualitative differences that we found between Bedford and Iceberg based on the dynamic OCT volumes that suggest a difference in resistance response (Fig. 2(e) and lines 106-123).

Minor comments:

1. In Fig. 2(b), it should be 0.3R+1.3G-0.8B inside the arrow, to be consistent with the context.

We updated this with the new optimized contrast

2. It is recommended to include color bars for all the intensity maps in Figs. 2 and 3.

We have updated the figures accordingly.

Reviewer #2 (Remarks to the Author):

Manuscript NCOMMS-23-64139 follows primarily the obligate pathogen B. lactucae in plant tissue without staining or fluorescent protein tags. This work demonstrates the potential utility in clarifying the disease cycles of these types of pathogens and in quantifying the level of resistance between cultivars. The methodology employed enabling 3D in vivo pathogen dynamics at increased resolution has potential application in visualizing molecular processes that occur during pathogen-host interactions.

Page 1, 2nd paragraph of introduction: The authors mention that measuring B. lactucae “enabled precise disease severity quantification”. It is difficult to show precision when the cultivars used are either resistant or susceptible with no in between. The results in Fig 2C show increase hyphal volume in cultivar Salinas, and significantly less so in both of the resistant cultivars. The gradation of level of resistance between the resistant cultivars was not resolved. This sentence should be rephrased to indicate correlation between hyphal volume of B. lactucae and resistance or susceptibility.

In our opinion we perform ‘precise disease quantification’ at the local level of the volume that we measure. Measuring multiple samples enabled quantifying a statistically significant difference between Salinas and Bedford and between Salinas and Iceberg. The reason that we did not obtain sufficient accuracy in resistance quantification to significantly distinguish Bedford and Iceberg is, we believe, not in the inaccuracy of the dOCT volume and length quantification, but in the small difference in resistance between the cultivars that makes it harder to obtain statistical significance.

To strengthen our claim of accuracy, we added additional DNA-based qPCR data of the *B. lactucae* infection of the same cultivars, Fig. 2(d). The results of the qPCR data is in line with the volume and length quantification, showing similar relative differences between the cultivars and ranking the resistance of the cultivars in the same order. This confirms our

hypothesis that the lack of significance is caused by the minor differences between Bedford and Iceberg resistance, and not by the inaccuracy of the method. Moreover, we highlight some qualitative dOCT features of the infection process that may be useful for qualitatively investigating differences in resistance responses (Fig. 2(e) and lines 106-123).

Exploring the accuracy of quantification with dynamic OCT, we added an additional analysis of the dependency of hyphal volume and length on the threshold (Supplementary section 2 and Supplementary Figure 5), which we discuss, together with the effect of the spatial resolution, in a separate paragraph in the discussion (lines 158-168).

Lastly, we rephrased the quoted sentence to “Through segmentation of *B. lactucae* hyphae in dOCT images we enable precise quantification of hyphal volume and length, which was in agreement with DNA-based quantification of whole leaf pathogen presence using qPCR” to keep our claim in line with the findings, and added a paragraph in the discussion on the accuracy and applicability of dOCT-based quantification for micro-phenotyping (lines 171-175).

Page 7, Methods: Is there a reference for B lactucae race Bl:33EU? Please insert. Also, what was the source of Bedford, Iceberg and Salinas from Figure 2? Couldn't find this information in the supplemental file either.

We added a reference to the press release of the identification of Bl:33EU, and also added the name of the company from where we acquired the seeds and *B. lactucae* material.

Page 3: What is “good contrast”? Perhaps contrast alone is a better choice.

We have rephrased this. However, good contrast is now quantified as shown by the color channel optimization scheme in the supplement Fig. 2.

Page 5, Figure 3: The authors indicate over 2 days in (a) but the labeling indicates 5d 18h, 4d 18h etc. Please correct, rewrite, or explain.

We started measuring at the end of the third day after inoculation because that gave the pathogen time to infect the leaf discs and gave a higher chance to image infection from the first time point. We clarified this in the text.

Page 5, Discussion: The phrase in the first sentence “something that has not been shown before” is not necessary.

We have removed the phrase.

Page 5, Discussion: Agree that a major benefit of the approach can be following infection in the same plant tissue over time. Would there be a way to adapt the approach be applied to attached leaves of a whole plant rather than detached leaf discs? Also, did the authors consider testing other lettuce tissue besides leaves, to determine if the colonization is systemic?

The method could be used for an attached leaf of small plants or seedlings when the sample is properly stabilized. Leaf infiltration will be more challenging but is certainly possible.

We have considered imaging *B. lactucae* infection in stems, but because the large majority of infection takes place in the leaves, we did not proceed imaging stems or root with infection. In other *B. lactucae* studies imaging stems or roots is seldom done, while leaf disc essays are more common. We added a paragraph in the discussion to address the application of our method on attached leaves, stem and roots. (lines 176-182)

Page 6, Methods: At bottom of page: what is a “good amount”? Perhaps sufficient amount is a better word choice.

We rephrased it to ‘most hyphae’.

Reviewer #3 (Remarks to the Author):

Dear authors,

*The article is well-written, illustrating the potential application of OCT in capturing the spatial and temporal dynamics of host-pathogen interactions, as seen with the fungus *Bremia lactucae*. The videos are quite illustrative.*

Thank you for your appreciation for our work.

*However, peculiarly, and lacking essential methodological details, the authors present findings involving downy mildew (*Hyaloperonospora arabidopsidis*) in *Arabidopsis* leaves and the root-knot nematode *Meloidogyne incognita* in pepper roots (crucial information such as acquisition methods, specific varieties involved, pathogen and parasitic nematode density, etc., is notably absent). Even new pathogen-host combinations are also added to the supplementary material, which doesn't make sense without a prior introductory note and without methodological details. To address this, either these segments should be omitted from the results and discussion, or more information need to be incorporated, including pertinent details, into the materials and methods section. Furthermore, a more in-depth analysis of the obtained results with all the pathogens/ plant parasitic nematodes is warranted. I would like to draw attention to the fact that, being a plant pathologist, I do not possess expertise in the more technical aspects. However, regarding the pathogen/parasite-host interaction, I couldn't help notice this omission in the manuscript*

In the revised manuscript, we have added a detailed explanation of the methodology so that the measurements are reproducible (Methods, lines 211-226). We would like to stress that, the purpose of the shown examples is not to draw significant biological conclusions, but to illustrate the wide applicability of dOCT by showing its ability to create images of other plant-pathogen systems with enhanced, label-free contrast. Therefore, an in-depth analysis of the results with pathogens other than *B. lactucae* is outside the scope of this manuscript and we limit ourselves to mentioning it briefly in the results section, and a short discussion in the supplementary materials.

Please, consider my suggestions throughout the text and highlight the necessary changes in the revised version. Also, it should be noted that the figures must be self-standing, and in this regard, all names should be written in full, with acronyms previously detailed.

We have adapted the manuscript accordingly.

REVIEWERS' COMMENTS

Reviewer #1 (Remarks to the Author):

The authors have addressed all my questions regarding the manuscript, and I am pleased with the revised version. I do not have any further major requests.

Please carefully check the typos that sparsely appear in the manuscript. For example, in Supplementary Table 2, line 5, "CP-value corrected volume" should be "P-value corrected volume".

Reviewer #2 (Remarks to the Author):

The authors have satisfactorily addressed all concerns of this reviewer in the revised ms.

Reviewer #3 (Remarks to the Author):

Dear authors,

Thank you for addressing most of my concerns and for revising the text accordingly. I believe the manuscript now more robustly reflects the work conducted.

I would like to draw your attention to the terminology used: it is important to differentiate between "infected" and "infested." Nematodes are parasites and, as such, are pests of plants, causing diseases. They infect the roots, but the soil is not infected; it is infested. Therefore, in the Methods section, line 217, please change "infected soil" to "infested soil."

Best wishes for your continued work.